# Oral Frailty and Its Relationship with Physical Frailty in Older Adults: A Longitudinal Study Using the Oral Frailty Five-Item Checklist

**DOI:** 10.3390/nu17010017

**Published:** 2024-12-24

**Authors:** Hiroshi Kusunoki, Yoko Hasegawa, Yasuyuki Nagasawa, Kensaku Shojima, Hiromitsu Yamazaki, Takara Mori, Shotaro Tsuji, Yosuke Wada, Kayoko Tamaki, Koutatsu Nagai, Ryota Matsuzawa, Hiromitsu Kishimoto, Hideo Shimizu, Ken Shinmura

**Affiliations:** 1Division of General Medicine, Department of Internal Medicine, Hyogo Medical University, Nishinomiya 663-8501, Hyogo, Japan; 2Department of Internal Medicine, Osaka Dental University, Hirakata 573-1121, Osaka, Japan; 3Division of Comprehensive Prosthodontics, Graduate School of Medical and Dental Sciences, Niigata University, Niigata 951-8514, Niigata, Japan; 4Department of Dentistry and Oral Surgery, Hyogo Medical University, Nishinomiya 663-8501, Hyogo, Japan; 5Amagasaki Medical COOP Honden Clinic, Amagasaki 660-0077, Hyogo, Japan; 6Department of Orthopedic Surgery, Hyogo Medical University, Nishinomiya 663-8501, Hyogo, Japan; 7School of Rehabilitation, Hyogo Medical University, Kobe 650-8530, Hyogo, Japan

**Keywords:** oral frailty, physical frailty, oral frailty five-item checklist

## Abstract

Background/Objectives: Oral frailty, first identified in Japan in 2014, refers to a state between healthy oral function and severe decline, marked by minor issues, such as tooth loss and chewing difficulties. The oral frailty five-item checklist (OF-5) enables non-dental professionals to evaluate oral frailty using five key indicators: remaining teeth count, chewing difficulties, swallowing difficulties, dry mouth, and articulatory oral skills. Limited studies exist. Methods: This study examined the relationship between oral and physical frailties in older adults and assessed the prognosis of physical frailty using the OF-5. Participants aged ≥65 years were recruited from the frail elderly in the Sasayama–Tamba area, Hyogo, Japan, and their physical function was assessed in terms of grip strength, walking speed, and skeletal muscle mass. Blood markers, such as cystatin C, an indicator of renal function, were also analyzed. Results: A cross-sectional analysis indicated that oral frailty was correlated with reduced muscle mass, walking speed, and physical function. Women had lower hemoglobin and albumin levels and a greater prevalence of frailty than men. Longitudinal analysis revealed that initial OF-5 scores predicted increased physical frailty after 2–3 years, especially in those with higher baseline scores. The OF-5 was a significant factor for frailty progression in both sexes. Conclusions: These results suggest that early detection of oral frailty via the OF-5 may be useful in preventing the progression of overall frailty in older adults.

## 1. Introduction

Oral health is a critical component of overall health, life satisfaction, quality of life, and self-perception. Impairment of oral function is highly prevalent among older adults, and aging has been reported to interact indirectly with various domains of frailty through multiple pathways. A clear example of this relationship is age-related functional oral deterioration, characterized by poor dental hygiene, inadequate dental prostheses, and dietary deficiencies, which collectively contribute to an increased risk of nutritional frailty [1].

Oral frailty is defined as an age-related gradual decline in oral function, often accompanied by deteriorations in physical functions. This condition is associated with significant adverse health outcomes in older adults, including increased mortality, physical frailty, functional disabilities, reduced quality of life, and a higher risk of hospitalization and falls [2]. Poor oral health in the elderly is a major health concern due to its links to the pathogenesis of systemic frailty, suggesting that it is a multidimensional geriatric syndrome. As such, oral frailty may serve as a potential risk factor for systemic frailty [3]. Oral frailty was first proposed by the Japanese Geriatrics Society in 2014, and is described as an age-related decrease in oral function. Oral frailty is further defined as “the overlap of minor declines in dental or oral functions that may increase the risk of adverse health outcomes” [4,5].

This condition poses an increased risk of further decline in oral function; however, it remains reversible if early and appropriate interventions are implemented. Signs of oral frailty, including decreased tongue pressure, increased food spillage, slight chewing difficulties, and a dry mouth, are often subtle and easily overlooked. Recent studies have shown that oral frailty in older adults not only affects oral health but also has systemic implications, contributing to overall frailty and sarcopenia (age-related muscle loss) [4,6,7].

In 2023, a new diagnostic criterion for oral frailty, known as the oral frailty five-item checklist (OF-5), was proposed [4]. It comprises five items: fewer teeth, difficulty in chewing, difficulty in swallowing, dry mouth, and low articulatory oral motor skills. The OF-5 is designed to be implemented in various settings beyond dental care facilities, including non-dental healthcare facilities and community activities, and can be assessed by older individuals. The OF-5 has demonstrated robust predictive validity for physical frailty, physical impairment, and mortality among the older population in Japan [4]. Despite these advancements, the longitudinal impact of oral frailty on the progression of physical frailty, as assessed by the OF-5, remains poorly understood, particularly in rural populations.

On 1 April 2024, the Japanese Geriatrics Society, Japanese Geriatric Dentistry Society, and Japanese Society for Sarcopenia and Frailty introduced a joint statement on oral frailty diagnosed via the OF-5 [8]. The OF-5 facilitates early detection of oral frailty and promotes interdisciplinary collaboration in its management, particularly in the medical and dental fields.

In our epidemiological study conducted among community-dwelling older adults in Sasayama–Tamba, Hyogo Prefecture (the frail elderly in the Sasayama–Tamba area [FESTA] study), we focused on the relationship between oral function and physical frailty. The rural environment of Sasayama–Tamba, which is relatively close to a metropolitan area and maintains a stable population without extreme depopulation or aging, offers a unique context. It features a modern, healthy, elderly population centered on suburban agriculture, with low population turnover. This setting provides an important backdrop for studying the interaction between oral and physical frailty given its distinctive demographic and health characteristics. In our previous study, we found a significant correlation between tongue pressure and cystatin C levels, an indicator of kidney function in the FESTA study. Our findings also revealed a correlation between tongue pressure, an indicator of oral function, and physical parameters, such as grip strength, walking speed, and muscle mass [9].

The Oral Frailty Checklist/Oral Frailty Index-8 (OFI-8) was developed by the Japan Dental Association [10,11], and consists of eight items: (1) difficulties in chewing; (2) difficulties in swallowing; (3) denture use; (4) dry mouth; (5) going out less frequently; (6) feasibility of chewing hard food; (7) brushing teeth at least twice a day; and (8) regular attendance at a dental clinic. Items (1) to (3) were scored as 2, whereas the other items were scored as 1. The maximum possible score was 11: low risk, 0–2 points; moderate risk, 3 points; and high risk, >4 points. Oral frailty, as assessed using the OFI-8, was independently associated with all-cause mortality, even after adjusting for physical and psychological frailty in older adults [12].

On the other hand, there are many reports on the associations of cystatin-C-related indices, including the creatinine-to-cystatin-C ratio (Cre/CysC ratio) and estimated glomerular filtration rate based on CysC (eGFRcys), with physical frailty and sarcopenia [13,14,15,16,17,18,19,20]. Our findings indicated that individuals at high risk for oral frailty, as assessed by the OFI-8, had lower levels of cystatin-C-related indices, grip strength, hemoglobin, and albumin, with a higher prevalence of oral frailty observed in women [21].

The OF-5 and OFI-8 share several common items, such as difficulties in chewing, difficulties in swallowing, and dry mouth. These three items are included in the Kihon checklist developed by the Japanese Ministry of Health, Labor, and Welfare, which consists of 25 questions in seven categories: physical strength, nutrition, eating, socialization, memory, mood, and lifestyle [22,23]. However, the OFI-5 differs from the OFI-8 in that it includes objective evaluations, such as the remaining teeth count and articulatory oral motor skills assessed by a dental specialist. The relationship between oral frailty and diagnosis using the OF-5, which includes objective measures based on dental examinations, grip strength, gait speed, and blood test indices, has not yet been examined. The comparative efficacy of the OF-5 in predicting physical frailty outcomes, especially in comparison to the OFI-8, has not been extensively explored, highlighting a vital area for investigation.

The longitudinal Kashiwa Study conducted by the University of Tokyo has also shown that oral frailty is a risk factor for physical frailty and is linked to life prognosis [3]. In the present study, we examined whether oral frailty, as diagnosed by the OF-5, predicts worsening physical frailty according to the Japanese Cardiovascular Health Study (J-CHS) criteria. Oral frailty, as assessed using the OF-5, has also been shown to be related to the development of physical disabilities and frailty [4].

This study aimed to assess, in a cross-sectional analysis, sex differences in physical and blood markers among individuals classified as having oral frailty by the OF-5. Additionally, using the OF-5, we aimed to investigate whether individuals classified as having suspected oral frailty (OF-5 score ≥ 2) exhibit differences in physical and biological markers, including height, weight, blood indices, and muscle strength, compared with those with lower OF-5 scores. We also explored whether these differences were associated with overall frailty. This study aimed to longitudinally assess the predictive value of the OF-5 checklist for physical frailty among older adults in Sasayama–Tamba by hypothesizing that higher OF-5 scores are significantly associated with an increased risk of physical frailty over time. Additionally, in a longitudinal analysis, we examined the association between OF-5 scores and the progression of physical frailty according to the J-CHS criteria. Finally, we evaluated the predictive value of OF-5 in comparison with other clinical markers over a follow-up period of 2–3 years.

## 2. Materials

### 2.1. Study Participants

This cross-sectional study within the FESTA study included individuals aged ≥65 years. Healthy community-dwelling older adults from the Sasayama–Tamba area, a rural region in Hyogo Prefecture, Japan, were recruited between 2017 and 2023. Body composition and blood sample analyses were performed as described previously [17,18]. Body composition was assessed using bioelectrical impedance analysis with an InBody 770 device (Inbody Japan Inc., Tokyo, Japan). Skeletal muscle mass index (SMI) was calculated as skeletal muscle mass divided by height squared (kg/m^2^). Handgrip strength was measured according to previously established methods [17,18,24].

This cross-sectional study included 313 men and 621 women (934 in total). For the longitudinal study, 105 men and 224 women (329 in total) from the first cross-sectional survey who had no missing data in the second survey conducted 2–3 years later were included.

All procedures performed in this study, which involved human participants, adhered to the ethical standards of the institutional and/or national research committee where the studies were conducted (IRB approval number Rinhi 0342 at Hyogo Medical University) and the 1964 Helsinki Declaration and its later amendments or comparable ethical standards.

### 2.2. Evaluation of Physical Function

To assess gait speed, the participants were instructed to walk a 12 m walkway at their usual pace, and the time taken to walk 10 m was recorded. Maximum grip strength was measured via a grip strength tester (GRIP-A; Takei Ltd., Niigata, Japan) [25]. Knee extension strength (Nm) of the dominant leg was evaluated during isometric contraction of the knee extensor in a seated position, with the knee maintained at a 60° angle using a hand-held dynamometer (Sakai Medical Co., Ltd., Tokyo, Japan) [26].

### 2.3. Diagnosis of Frailty

Frailty phenotypes were assessed based on the five clinical features defined in the Cardiovascular Health Study (CHS): slow gait speed, weakness, exhaustion, low physical activity, and weight loss [27]. The frailty score was calculated using a modified version of the CHS (J-CHS) [28]. The number of applicable frailty phenotypes of the five was used to determine the J-CHS score. A score of 0 was defined as robust, 1 or 2 as pre-frail, and ≥3 as frail.

In a longitudinal study, during the second survey conducted 2–3 years after the first survey, the participants were categorized based on changes in their J-CHS frailty scores. Seventy-four participants (27 men and 47 women) whose scores had increased were defined as “worsened”, 167 (51 men, 116 women) whose scores remained unchanged were categorized as “unchanged”, and 88 (27 men, 61 women) whose scores had decreased were classified as “improved”. Changes in the J-CHS frailty scores were used to classify the participants as improved, unchanged, or worsened, and comparisons were made across groups in terms of physical indices, blood markers, OF-5 scores, and the number of positive subjects for each item at the time of the first survey. Logistic regression analysis, including other indices, was used to determine whether the J-CHS scores worsened during the second survey.

### 2.4. Diagnosis of Sarcopenia

Sarcopenia was defined according to the criteria for the Asia Working Group for Sarcopenia (AWGS) 2019 [29]. Body composition was evaluated by bioelectrical impedance analysis (BIA) using an InBody 770^®^ (InBody Japan Inc., Tokyo, Japan). The skeletal muscle mass index (SMI) was calculated as SMM/height^2^ (kg/m^2^). The handgrip power, the normal and maximal gait speed, five-time chair stand test (5CS), Timed Up and Go test (TUG), and Short Physical Performance Battery (SPPB) scores were evaluated as described previously [29]. Sarcopenia was considered if the participants had a low SMI (<7.0 kg/m^2^ in men; <5.7 kg/m^2^ in women) and weak handgrip strength (<28 kg in men; <18 kg in women) or low physical performance (normal gait speed < 1.0 m/s, 5CS ≥ 12 s or SPPB ≤ 9).

### 2.5. Evaluation of Oral Function

The participants were seated in reclinable nursing chairs and underwent oral examinations. The number of remaining teeth, occlusal force, and tongue pressure were assessed. Tongue pressure was measured twice using a JMS Tongue Pressure Measuring Device (JMS Co., Ltd., Hiroshima, Japan), and the highest value was recorded [30]. To evaluate tongue motor function (oral diadochokinesis [ODK]), we used oral function measurement equipment (KENKOU-KUN Handy; Takei Scientific Instruments Co., Ltd., Niigata, Japan) to measure the articulatory velocity of /ta/ [31].

### 2.6. Calculation of eGFR

We calculated creatine-based eGFR (eGFRcre) and eGFRcys using equations provided by the Japanese Society of Nephrology [32,33].

### 2.7. Statistical Analysis

The results are expressed as the means ± standard deviations or percentages. For intergroup comparisons, Student’s *t*-test was used for data analysis. Categorical variables are presented as absolute numbers (n) and relative frequencies (%), and were analyzed using the Fisher’s exact test. Univariate and multivariate logistic regression analyses were performed to calculate the odds ratios and 95% confidence intervals. Data analysis was conducted using JMP version 17.1 software, with statistical significance set at *p* < 0.05.

## 3. Results

The characteristics of 313 men and 621 women (934 in total) are shown in Table 1. The prevalence of oral frailty was slightly >40% in both sexes, with no significant difference between the sexes. According to the J-CHS frailty criteria by sex, exhaustion tended to be greater in women than in men; however, there were no significant differences in the other four criteria. Muscle strength, muscle mass, and walking speed were generally greater in men than in women, although there were no sex differences in the Timed Up and Go (TUG) test or the five-time chair stand (5CS) test. Tongue pressure was greater in men, whereas ODK tended to be greater in women. However, no significant sex-related differences were observed in the number of teeth. Women also tended to be more prone to anemia, with higher total protein and albumin levels. Creatine, cystatin C, and Cre/CysC levels reflected muscle mass and tended to be higher in men than in women, whereas eGFR tended to be higher in women than in men.

Using the J-CHS criteria for the diagnosis of physical frailty, more than half of the participants in both men and women were categorized as prefrail, while 4.4% of the overall population were diagnosed as frail (3.2% in men and 5.0% in women). A typical phenotype of physical frailty is age-related muscle loss, known as sarcopenia. Using the AWGS2019 diagnostic criteria for Asians, the prevalence of sarcopenia was estimated to be approximately 7% in both men and women, with no significant gender differences.

A score of ≥2 on the OF-5 indicates a diagnosis of oral frailty, which is associated with older age, shorter height, lower muscle mass and strength, and reduced physical functions, such as walking speed, TUG, and the 5CS test. Tongue pressure, number of teeth, and ODK were also reduced in individuals with oral frailty. The prevalence of sarcopenia was observed to be higher in both men and women with oral frailty. However, the difference was not statistically significant in men. Overall, the findings suggest that sarcopenia exhibits a similar trend to physical frailty (Table 2).

Cystatin-C-related indices, including the Cre/CysC ratio and eGFRcys, which we have previously reported, were lower in individuals of both sexes with oral frailty [21]. Additionally, hemoglobin and albumin levels were lower in women with oral frailty. Both men and women with oral frailty were less robust and had more pre-frailty; however, owing to the small number of frail individuals, the difference was not significant in the amount of frailty.

A longitudinal study involving 329 participants (105 men and 224 women) revealed changes in oral function between the first and second follow-up surveys. Two to three years passed between the first and second follow-ups, during which no significant changes were observed in body size, grip strength, walking speed, SMI, or other parameters. However, cystatin-C-related indices (Cre/CysC, eGFRcys, and eGFRcys/eGFRcre) were significantly lower at the second follow-up in both men and women. In the second follow-up survey, albumin levels and tongue pressure did not significantly decrease in men but did show a significant decrease in women. According to the J-CHS frailty criteria, there was a tendency for a decrease in pre-frailty among women during the second follow-up. Approximately half of the participants, both men and women, showed no changes in the relevant J-CHS items, whereas approximately a quarter showed either improvement or worsening (Table 3).

We also analyzed the baseline characteristics of the groups classified as improved/unchanged and worsened. In men, lower grip strength and fewer teeth at baseline were associated with disease worsening. In those who worsened, an OF-5 score of ≥2 at baseline was common, and many patients were assessed as having oral frailty at the first time point. Among women, swallowing and chewing problems were more frequently reported at baseline in the worsened group, although the only significant sex difference was observed in the total OF-5 scores. In summary, individuals with higher baseline OF-5 scores were more likely to experience worsening J-CHS scores at the second time point (Table 4).

Univariate logistic regression analysis was conducted to calculate the odds ratios for each indicator at baseline in the worsening group in the second survey. For men, significant associations were found between reduced grip strength and tooth loss, whereas an OF-5 score of ≥2 and a diagnosis of oral frailty at the first visit were also significant worsening risk factors. Significant associations were observed between decreased chewing ability and swallowing function in the women. The OF-5 score was a significant worsening risk factor in both men and women; however, in women, there was no significant difference in those with an OF-5 score ≥2 (Table 5A).

A multivariate logistic regression analysis was performed using age, BMI, grip strength, gait speed, SMI, Cre/CysC ratio, and OF-5 score, which are associated with frailty and sarcopenia, as explanatory variables. In men, grip strength was identified as a significant risk factor in univariate analysis; consequently, OF-5 score did not remain a significant risk factor in multivariate analysis. However, in women, the OF-5 score remained a significant risk factor (Table 5B).

The same multivariate logistic regression analysis was repeated for men by adjusting the OF-5 score to ≥2, and a significant difference remained (Table 5C).

## 4. Discussion

This study offers a comprehensive examination of sex-specific differences in oral frailty and related factors in a cohort of older adults, highlighting the important aspects of oral and physical function. Oral dysfunction is regarded as a significant contributor to systemic decline. Oral frailty is defined as a mild decline in oral functions during the early and reversible stages of frailty. Many community-dwelling older people have reduced oral function or oral hypofunction, which is significantly associated with frailty and aging. Appropriate evaluation of oral function and effective intervention to suppress oral function deterioration may be effective in extending the healthy life expectancy of older people [34].

Frailty, in contrast, is considered a state of increased vulnerability to disease onset and physical dysfunction due to a decline in several functions associated with aging. Sarcopenia, a state of reduced muscle mass, is a typical physical frailty phenotype. The prevalence of oral frailty is similar between men and women, affecting >40% of the population. The 40% prevalence of oral frailty was in agreement with several previous reports [4,35,36].

Women tended to report more fatigue and anemia, whereas men reported greater muscle strength, muscle mass, and tongue pressure. These findings underscore the need to consider sex-based physiological differences when evaluating frailty and sarcopenia, particularly oral-health-related parameters.

One of the key results of this study was the significant association between oral frailty and reduced physical functions, such as walking speed, muscle mass, and tongue pressure, confirming an intricate link between systemic frailty and oral health. In both sexes, a higher OF-5 score of ≥2, which indicates a diagnosis of oral frailty, was correlated with diminished physical and oral functions, including grip strength and the number of teeth. These findings suggest that oral frailty can serve as a valuable early marker of declining physical capacity and could help identify individuals at risk of sarcopenia or broader systemic frailty.

The longitudinal component of this study provides further insights into the progression of oral frailty. Over a follow-up period of 2–3 years, significant declines in cystatin-C-related indices (Cre/CysC and eGFRcys) and oral functions, including tongue pressure and albumin levels, were observed, particularly in women. These changes were not accompanied by significant alterations in muscle strength, walking speed, or other systemic parameters, indicating that oral frailty may progress rapidly or independently of systemic physical decline. This underlines the importance of targeted interventions focusing on oral health to mitigate the progression of frailty.

Importantly, the logistic regression analysis identified distinct risk factors for the worsening of oral frailty. In men, reduced grip strength and tooth loss were significant predictors, consistent with previous studies linking oral health to systemic physical capacity. In contrast, impaired chewing and swallowing functions were more prominent risk factors in women, underscoring the role of oral function in overall health deterioration in older women. Notably, although the OF-5 score was a significant risk factor for worsening frailty in both sexes, its effect was more pronounced in women, suggesting a potential sex difference in the relationship between oral health and progression of systemic frailty.

Approximately one-fourth of the participants demonstrated improvements in J-CHS scores for both men and women during the second survey. Neither pharmacological nor exercise interventions specifically targeting frailty were implemented. Therefore, the observed improvement in J-CHS scores may be attributed to increased activity levels among older adults following the lifting of COVID-19-related restrictions in Japan, such as the state of emergency declarations.

This interpretation is supported by the findings in Table 3, which show a decrease in the number of participants categorized under ’Low Activity’ during the second survey for both men and women. During the first survey, many older adults had limited outdoor activities due to self-imposed restrictions stemming from the pandemic. In contrast, by the second survey, a substantial number of these older adults had resumed their regular activities, potentially explaining the observed changes.

Despite these significant findings, it is important to acknowledge the limitations of this study. The sample size of frail individuals was relatively small, limiting the ability to detect subtle differences between sexes or within subgroups. Additionally, although longitudinal data were collected, the follow-up period may not have been sufficiently long to fully capture the trajectory of oral frailty in this population. Future studies with larger, more diverse cohorts and extended follow-up periods are needed to clarify the dynamics between oral and systemic frailty, and to identify effective interventions that target both domains. Moreover, the inclusion of healthy volunteers may have influenced the representativeness of the study population. Previous reports indicated that the prevalence of frailty, according to the J-CHS criteria, is approximately 10% among older adults in the Japanese populations [37,38,39]. However, in this study, the prevalence of frailty was considerably lower at 3.2% for men and 5.0% for women.

During the 2–3-year observation period, no significant decline in grip strength, gait speed, or muscle mass was observed. However, even with a limited number of participants and short observation period, a diagnosis of oral frailty using the OF-5 was associated with an increase in J-CHS scores, suggesting that the OF-5 score is significantly linked to the worsening of long-term physical frailty. Oral frailty, as assessed using the OF-5, was also significantly associated with higher J-CHS scores after adjusting for age, BMI, grip strength, gait speed, SMI, and other frailty-related factors. These findings indicate that the OF-5 is a promising predictor of frailty onset. The novelty of this study lies in the significant relationship between OF-5 score and other frailty indices.

In the original article that introduced the OF-5 in 2023, difficulty in chewing, difficulty swallowing, and dry mouth were evaluated using subjective questionnaires, whereas objective data from dental examinations were used to assess the number of teeth and articulatory oral motor skills. Similarly, the present study evaluated articulatory oral-motor skills via ODK, and a dentist assessed the number of teeth, to objectively evaluate these aspects.

The results of the objective evaluation of ODK and subjective evaluation via questionnaires were in good agreement [39]. In April 2024, a joint consensus statement on “Oral Frailty” in Japan suggested that the objective assessment of ODK is no longer necessary and can be replaced with the following question: “Have you had difficulty with clear pronunciation recently?”. This statement also allows for a self-reported assessment of whether respondents had >20 teeth. Future studies should investigate whether oral frailty, as assessed by the OF-5 subjective questionnaire, is associated with lower muscle mass, slower gait speed, and reduced physical function in cross-sectional studies, and whether it significantly correlates with the progression of physical frailty in longitudinal studies.

## 5. Conclusions

In conclusion, this study provides valuable evidence on the relationships between oral frailty, systemic frailty, and risk factors in older adults. These findings emphasize the importance of integrating oral health assessments into frailty screening protocols, particularly for older women in whom oral dysfunction may serve as an early marker of systemic health decline. These insights have implications for the development of interventions aimed at preventing or mitigating frailty and its associated adverse outcomes in aging populations.

## Figures and Tables

**Table 1 nutrients-17-00017-t001:** Baseline characteristics of participants according to sex.

	Total (n = 934)	Men (n = 313)	Women (n = 621)	*p*
Age (years)	74.0 ± 5.8	74.7 ± 5.9	73.6 ± 5.8	**<0.001**
Height (cm)	155.4 ± 8.3	163.9 ± 6.0	151.1 ± 5.5	**<0.001**
Body weight (kg)	55.0 ± 9.4	62.5 ± 8.8	51.2 ± 7.1	**<0.001**
Body mass index	22.7 ± 2.9	23.2 ± 2.8	22.4 ± 2.9	**<0.001**
Skeletal muscle mass (SMM) (kg)	15.7 ± 3.7	19.9 ± 2.8	13.6 ± 1.9	**<0.001**
Skeletal muscle mass index (SMI)	6.43 ± 0.93	7.37 ± 0.73	5.95 ± 0.59	**<0.001**
Body fat mass (kg)	15.3 ± 5.3	14.8 ± 5.3	15.6 ± 5.3	**0.035**
Percentage of BFM (%)	27.6 ± 7.3	23.2 ± 6.1	29.8 ± 6.9	**<0.001**
Grip power (kg)	26.7 ± 7.6	34.6 ± 6.5	22.7 ± 4.2	**<0.001**
Knee extension muscle strength (Nm)	336.5 ± 115.2	428.7 ± 114.2	290.0 ± 83.4	**<0.001**
Normal gait speed (m/s)	1.41 ± 0.24	1.38 ± 0.24	1.43 ± 0.24	**<0.001**
Maximal gait speed (m/s)	1.87 ± 0.32	1.94 ± 0.34	1.84 ± 0.30	**<0.001**
Timed Up and Go test (TUG)	6.33 ± 1.64	6.30 ± 1.93	6.35 ± 1.47	0.675
Five-time chair stand test (5CS)	7.53 ± 4.15	7.90 ± 4.77	7.34 ± 3.79	0.137
Cre (mg/dL)	0.75 ± 0.28	0.90 ± 0.19	0.68 ± 0.29	**<0.001**
CysC (mg/L)	0.95 ± 0.25	1.03 ± 0.23	0.91 ± 0.24	**<0.001**
Cre/CysC	0.80 ± 0.13	0.89 ± 0.12	0.75 ± 0.11	**<0.001**
eGFRcre (mL/min/1.73 m^2^)	67.0 ± 14.1	66.2 ± 14.0	67.5 ± 14.1	**<0.001**
eGFRcys (mL/min/1.73 m^2^)	73.3 ± 16.1	70.6 ± 16.5	74.7 ± 15.8	**<0.001**
eGFRcys/eGFRcre	1.10 ± 0.17	1.07 ± 0.17	1.12 ± 0.17	**<0.001**
Red blood cell (×104/μL)	439.5 ± 43.8	452.7 ± 48.5	432.9 ± 39.6	**<0.001**
Hemoglobin (g/dL)	13.5 ± 1.3	14.2 ± 1.5	13.2 ± 1.1	**<0.001**
Hematocrit (%)	40.8 ± 3.7	42.4 ± 4.0	40.1 ± 3.2	**<0.001**
Total protein (g/dL)	7.35 ± 0.47	7.33 ± 0.44	7.37 ± 0.39	**<0.001**
Albumin (g/dL)	4.32 ± 0.32	4.29 ± 0.31	4.34 ± 0.27	**<0.001**
Number of teeth, n	20.1 ± 8.8	19.6 ± 9.4	20.3 ± 8.5	0.357
Tongue pressure (kPa)	33.5 ± 8.6	34.3 ± 9.0	33.1 ± 8.3	**0.038**
Low articulatory oral motor skills (times/s)	6.05 ± 0.97	5.84 ± 1.12	6.16 ± 0.87	**<0.001**
**Item of oral frailty**				
Fewer teeth	316(33.8)	113(36.1)	203(32.7)	0.306
Difficulty in chewing	185(19.8)	51(16.3)	134(21.6)	0.056
Difficulty in swallowing	241(25.8)	65(20.8)	176(28.3)	**0.014**
Dry mouth	288(30.8)	73(23.2)	215(34.6)	**<0.001**
Low articulatory oral motor skills	336(36.0)	135(43.1)	201(32.4)	**0.002**
**OF-5 score**	1.46 ± 1.20	1.40 ± 1.11	1.50 ± 1.24	0.482
0, n(%)	227(24.3)	77(24.6)	150(24.2)	0.872
1, n(%)	296(31.7)	97(31.0)	199(32.0)	0.766
2, n(%)	234(25.0)	92(29.4)	142(22.9)	**0.031**
3, n(%)	117(12.6)	33(10.8)	84(13.5)	0.210
4, n(%)	49(5.2)	13(4.2)	36(5.8)	0.352
5, n(%)	11(1.2)	1(0.3)	10(1.6)	0.111
**Oral frailty status**				
Oral non-frailty, 0–1 OF-5 score, n(%)	523(56.0)	174(55.6)	349(56.2)	0.889
Oral frailty, ≥2 OF-5 score, n(%)	411(44.0)	139(44.4)	272(43.8)	
**Item of frailty (J-CHS)**				
Shrinking, n(%)	140(15.0)	47(15.0)	93(15.0)	1.000
Weakness (grip strength < 28 kg in men or 18 kg in women), n(%)	95(10.1)	25(8.0)	70(11.3)	0.136
Exhaustion, n(%)	223(23.9)	61(19.5)	162(26.0)	**0.028**
Slowness (gait speed < 1.0 m/s), n(%)	41(4.4)	14(4.5)	27(4.3)	1.000
Low activity, n(%)	261(27.9)	97(31.0)	164(26.4)	0.143
**Frailty status (J-CHS criteria)**				
Robust, n(%)	405(43.4)	138(44.1)	267(43.0)	0.780
Pre-frailty, n(%)	488(52.2)	165(52.7)	323(52.0)	0.890
Frailty, n(%)	41(4.4)	10(3.2)	31(5.0)	0.239
**Sarcopenia status (AWGS criteria)**				
Sarcopenia	67(7.2)	22(7.0)	45(7.2)	1.000
Non-sarcopenia	867(92.8)	291(93.0)	576(92.8)	

Cre, creatinine; CysC, cystatin C; eGFRcys, cystatin-based estimated glomerular filtration rate; eGFRcre, creatinine-based estimated glomerular filtration rate; OF-5, oral frailty five-item checklist; J-CHS, Japanese Cardiovascular Health Study criteria. The *p*-value represents the significant difference between women and men.

**Table 2 nutrients-17-00017-t002:** Comparison of physical and oral function according to oral frailty status in men and women.

	Men (n = 313)			Women (n = 621)		
	Oral Non-Frailty, 0–1 OF-5 Score (n = 174)	Oral Frailty, ≥2 OF-5 Score (n = 139)	*p*	Oral Non-Frailty, 0–1 OF-5 Score (n = 349)	Oral Frailty, ≥2 OF-5 Score (n = 272)	*p*
**Age (years)**	73.7 ± 5.4	76.0 ± 6.2	**<0.001**	72.1 ± 5.3	75.5 ± 5.9	**<0.001**
**Height (cm)**	164.6 ± 5.9	163.1 ± 6.2	**0.018**	151.8 ± 5.3	150.2 ± 5.6	**<0.001**
**Body weight (kg)**	63.2 ± 8.6	61.6 ± 9.1	0.107	51.6 ± 7.4	50.6 ± 6.7	0.103
**Body mass index**	23.3 ± 2.6	23.2 ± 3.1	0.715	22.4 ± 2.9	22.5 ± 3.0	0.639
**Skeletal muscle mass (SMM) (kg)**	20.3 ± 2.8	19.3 ± 2.7	**0.001**	13.9 ± 1.9	13.3 ± 2.0	**<0.001**
**Skeletal muscle mass index (SMI)**	7.48 ± 0.71	7.24 ± 0.74	**0.005**	6.00 ± 0.58	5.88 ± 0.59	**0.014**
**Body fat mass (kg)**	14.6 ± 4.9	15.1 ± 5.7	0.453	15.7 ± 5.5	15.6 ± 5.1	0.894
**Percentage of BFM (%)**	22.7 ± 5.6	23.8 ± 6.6	0.095	29.6 ± 7.0	30.1 ± 6.8	0.339
**Grip power (kg)**	35.7 ± 6.3	33.2 ± 6.5	**<0.001**	23.5 ± 4.2	21.7 ± 4.1	**<0.001**
**Knee extension muscle strength (Nm)**	446.1 ± 110.9	406.9 ± 115.3	**0.002**	304.0 ± 74.1	271.7 ± 90.6	**<0.001**
**Normal gait speed (m/s)**	1.42 ± 0.24	1.35 ± 0.24	**0.007**	1.47 ± 0.23	1.38 ± 0.24	**<0.001**
**Maximal gait speed (m/s)**	2.00 ± 0.32	1.86 ± 0.34	**<0.001**	1.90 ± 0.28	1.76 ± 0.31	**<0.001**
**Timed Up and Go test (TUG)**	6.02 ± 1.64	6.65 ± 2.19	**0.004**	6.03 ± 1.21	6.75 ± 1.68	**<0.001**
**Five-time chair stand test (5CS)**	7.25 ± 2.49	8.71 ± 6.51	**0.007**	6.90 ± 3.36	7.91 ± 4.21	**<0.001**
**Cre (mg/dL)**	0.91 ± 0.21	0.88 ± 0.19	0.203	0.66 ± 0.13	0.68 ± 0.14	0.931
**CysC (mg/L)**	1.01 ± 0.23	1.04 ± 0.25	0.134	0.89 ± 0.28	0.93 ± 0.19	0.042
**Cre/CysC**	0.91 ± 0.12	0.86 ± 0.13	**<0.001**	0.76 ± 0.11	0.73 ± 0.10	**0.001**
**eGFRcre (mL/min/1.73 m^2^)**	65.8 ± 14.5	66.7 ± 13.3	0.562	68.3 ± 13.9	66.3 ± 14.2	0.085
**eGFRcys (mL/min/1.73 m^2^)**	72.2 ± 16.5	68.5 ± 16.3	**0.049**	77.1 ± 15.6	71.7 ± 15.4	<0.001
**eGFRcys/eGFRcre**	1.11 ± 0.16	1.03 ± 0.18	**<0.001**	1.14 ± 0.17	1.09 ± 0.17	<0.001
**Red blood cell (×10^4^/μL)**	453.3 ± 47.9	452.0 ± 49.4	0.823	436.2 ± 37.2	428.7 ± 42.1	0.020
**Hemoglobin (g/dL)**	14.2 ± 1.4	14.2 ± 1.6	0.826	13.3 ± 1.1	13.1 ± 1.1	0.012
**Hematocrit (%)**	42.4 ± 3.7	42.4 ± 4.3	0.968	40.3 ± 3.1	39.8 ± 3.3	**0.041**
**Total protein (g/dL)**	7.28 ± 0.43	7.36 ± 0.77	**0.029**	7.39 ± 0.39	7.35 ± 0.39	0.189
**Albumin (g/dL)**	4.28 ± 0.31	4.29 ± 0.48	0.286	4.36 ± 0.25	4.31 ± 0.29	**0.017**
**Tongue pressure (kPa)**	35.3 ± 8.4	33.1 ± 9.6	**0.032**	33.7 ± 7.5	32.4 ± 9.1	0.066
**Number of teeth, n**	23.0 ± 7.5	15.3 ± 10.0	**<0.001**	23.4 ± 6.5	16.4 ± 9.1	**<0.001**
**Fewer teeth**	27(15.8)	87(62.1)		48(13.8)	155(57.0)	
**Low articulatory oral motor skills (times/s)**	6.21 ± 0.88	5.37 ± 1.20	**<0.001**	6.22 ± 0.74	5.82 ± 0.92	**<0.001**
**Low articulatory oral motor skills**	37(21.6)	99(70.7)	**<0.001**	56(16.0)	145(53.3)	**<0.001**
**Item of frailty (J-CHS criteria)**						
Shrinking	23(13.2)	24(17.3)	0.342	40(11.4)	53(19.4)	**0.006**
Weakness (Grip strength < 28 kg in men or 18 kg in women), n(%)	9(5.2)	16(11.5)	0.057	24(6.9)	46(16.9)	**<0.001**
Exhaustion, n(%)	19(10.9)	42(30.2)	**<0.001**	64(18.3)	98(36.0)	**<0.001**
Slowness (Gait speed < 1.0 m/s), n(%)	6(3.4)	8(5.8)	0.412	11(3.2)	16(5.9)	0.114
Low activity, n(%)	47(27.0)	50(31.0)	0.110	64(18.3)	71(26.1)	**0.024**
**Frailty status (J-CHS criteria)**						
Robust, n(%)	92(52.9)	46(33.1)	**<0.001**	91(33.5)	176(50.4)	**<0.001**
Pre-frailty, n(%)	79(45.4)	86(61.9)	**0.004**	159(58.5)	164(47.0)	**<0.001**
Frailty, n(%)	3(1.7)	7(5.0)	0.115	22(8.0)	9(2.6)	0.097
**Sarcopenia status (AWGS criteria)**						
Sarcopenia, n(%)	6(3.4)	16(11.5)	0.120	17(4.9)	28(10.3)	**0.012**
Non-sarcopenia, n(%)	94(96.6)	123(88.5)		332(95.1)	244(89.7)	

Cre, creatinine; CysC, cystatin C; eGFRcys, cystatin-based estimated glomerular filtration rate; eGFRcre, creatinine-based estimated glomerular filtration rate; OF-5, oral frailty five-item checklist; J-CHS, Japanese Cardiovascular Health Study criteria.

**Table 3 nutrients-17-00017-t003:** Changes in physical and oral function from baseline to follow-up according to sex.

	Men (n = 105)			Women (n = 224)		
	First Survey	Second Survey	*p*	First Survey	Second Survey	*p*
**Number of days to second survey**	955.1 ± 351.7			985.5 ± 348.0		
**Age (years)**	73.6 ± 5.8	76.2 ± 5.9	**0.001**	72.1 ± 5.4	74.8 ± 5.4	**<0.001**
**Height (cm)**	164.1 ± 6.5	163.6 ± 6.6	0.566	151.4 ± 5.2	150.8 ± 5.3	0.252
**Body weight (kg)**	63.2 ± 8.5	62.3 ± 9.0	0.440	51.4 ± 6.6	51.1 ± 6.8	0.618
**Body mass index**	23.4 ± 2.6	23.2 ± 2.8	0.585	22.5 ± 2.8	22.5 ± 2.9	0.912
**Skeletal muscle mass index (SMI)**	7.44 ± 0.67	7.33 ± 0.70	0.242	5.93 ± 0.56	5.89 ± 0.58	0.397
**Grip power (kg)**	35.8 ± 7.1	34.2 ± 6.7	0.087	23.2 ± 4.2	22.8 ± 3.8	0.311
**Normal gait speed (m/s)**	1.39 ± 0.23	1.35 ± 0.25	0.249	1.44 ± 0.23	1.40 ± 0.23	0.117
**Cre (mg/dL)**	0.91 ± 0.19	0.92 ± 0.23	0.564	0.66 ± 0.14	0.69 ± 0.17	0.140
**CysC (mg/L)**	1.00 ± 0.22	1.09 ± 0.30	**0.017**	0.89 ± 0.17	0.95 ± 0.23	**0.001**
**Cre/CysC**	0.92 ± 0.13	0.86 ± 0.12	**0.001**	0.75 ± 0.10	0.72 ± 0.10	**0.005**
**eGFRcre (mL/min/1.73 m^2^)**	65.8 ± 13.5	64.4 ± 13.5	0.453	68.6 ± 13.3	66.2 ± 13.4	0.052
**eGFRcys (mL/min/1.73 m^2^)**	72.7 ± 15.6	66.2 ± 15.3	**0.003**	76.2 ± 14.6	70.3 ± 13.9	**<0.001**
**eGFRcys/eGFRcre**	1.12 ± 0.18	1.03 ± 0.16	**<0.001**	1.12 ± 0.17	1.07 ± 0.16	**0.002**
**Hemoglobin (g/dL)**	14.2 ± 1.4	14.0 ± 1.4	0.227	13.2 ± 1.0	13.1 ± 1.0	0.087
**Albumin (g/dL)**	4.3 ± 0.3	4.2 ± 0.3	0.128	4.4 ± 0.3	4.3 ± 0.3	**<0.001**
**Number of teeth, n**	20.7 ± 8.8	19.6 ± 8.8	0.380	21.1 ± 8.2	20.2 ± 8.3	0.271
**Tongue pressure (kPa)**	34.8 ± 8.8	34.4 ± 9.2	0.734	33.4 ± 8.5	31.8 ± 8.5	**0.040**
**Low articulatory oral motor skills (times/s)**	5.83 ± 1.12	5.98 ± 1.08	0.329	6.18 ± 0.80	6.13 ± 0.94	0.575
**Item of oral frailty**						
Fewer teeth, n(%)	36(34.3)	40(38.1)	0.667	67(29.9)	74(33.0)	0.542
Difficulty in chewing, n(%)	18(17.1)	26(24.8)	0.235	42(18.8)	47(21.0)	0.542
Difficulty in swallowing, n(%)	24(22.9)	29(27.6)	0.525	63(28.1)	61(27.2)	0.636
Dry mouth, n(%)	25(23.8)	30(28.6)	0.530	70(31.3)	73(32.7)	0.916
Low articulatory oral motor skills, n(%)	47(44.8)	35(33.3)	0.120	74(33.0)	76(33.9)	0.762
**OF-5 score**	1.43 ± 1.03	1.52 ± 1.13	0.523	1.41 ± 1.22	1.48 ± 1.20	0.559
0, n(%)	21(20.0)	21(20.0)	1.000	62(27.7)	50(22.3)	0.230
1, n(%)	36(34.3)	33(31.4)	0.663	69(30.8)	79(35.3)	0.318
2, n(%)	33(31.4)	32(30.5)	1.000	47(21.0)	50(22.3)	0.819
3, n(%)	12(11.4)	14(13.3)	0.835	33(14.7)	32(14.3)	1.000
4, n(%)	3(2.9)	4(3.8)	1.000	11(4.9)	9(4.0)	0.820
5, n(%)	0	1(1.0)	1.000	2(0.9)	4(1.8)	0.685
**Oral frailty states**						
Oral non-frailty, 0–1 OF-5 score, n(%)	57(54.9)	54(51.4)	0.782	131(58.7)	129(57.8)	0.849
Oral frailty, ≥2 OF-5 score, n(%)	48(45.7)	51(48.5)		92(41.3)	94(42.2)	
**Item of frailty (J-CHS criteria)**						
Shrinking	17(16.2)	17(16.2)	1.000	35(15.6)	33(14.7)	0.895
Weakness (grip strength < 28 kg in men or 18 kg in women)	7(6.7)	15(14.3)	0.113	16(7.2)	20(8.9)	0.603
Exhaustion	18(17.1)	22(21.0)	0.598	49(21.9)	56(25.0)	0.504
Slowness (gait speed < 1.0 m/s)	4(3.8)	8(7.6)	0.373	6(2.7)	11(4.9)	0.323
Low activity	36(34.3)	23(21.9)	0.065	74(33.0)	41(18.3)	**<0.001**
**J-CHS frailty status**						
Robust, n(%)	43(41.0)	45(42.9)	0.889	92(41.1)	111(49.6)	0.088
Pre-frailty, n(%)	60(57.1)	58(55.2)	0.889	125(55.8)	99(44.2)	**0.014**
Frailty, n(%)	2(1.9)	2(1.9)	1.000	7(3.1)	14(6.2)	0.179
**J-CHS change category**						
Improved, n(%)	27(25.7)			61(27.2)		
Unchanged, n(%)	51(48.6)			116(51.8)		
Worsened, n(%)	27(25.7)			47(21.0)		

Cre, creatinine; CysC, cystatin C; eGFRcys, cystatin-based estimated glomerular filtration rate; eGFRcre, creatinine-based estimated glomerular filtration rate; OF-5, oral frailty five-item checklist; J-CHS, Japanese Cardiovascular Health Study criteria.

**Table 4 nutrients-17-00017-t004:** Baseline characteristics and oral frailty according to frailty progression status in men and women.

Results of First Survey	Men (n = 105)			Women (n = 224)		
	Improved or Unchanged (n = 78)	Worsened (n = 27)	*p*	Improved or Unchanged (n = 177)	Worsened (n = 47)	*p*
Age (years)	73.1 ± 5.7	75.1 ± 5.9	0.108	71.9 ± 5.4	73.2 ± 5.1	0.125
Height (cm)	164.3 ± 6.6	163.4 ± 6.2	0.544	151.5 ± 5.1	150.8 ± 5.6	0.426
Body weight (kg)	63.1 ± 8.3	63.6 ± 9.1	0.788	51.2 ± 6.5	52.4 ± 6.8	0.250
Body mass index	23.3 ± 2.5	23.8 ± 3.0	0.452	22.3 ± 2.8	23.0 ± 2.8	0.106
Skeletal muscle mass index (SMI)	7.48 ± 0.65	7.32 ± 0.70	0.276	5.93 ± 0.54	5.95 ± 0.67	0.784
Grip power (kg)	36.7 ± 7.4	33.2 ± 5.6	**0.027**	1.45 ± 0.22	1.44 ± 0.24	0.810
Normal gait speed (m/s)	1.41 ± 0.23	1.34 ± 0.22	0.215	23.3 ± 4.3	22.5 ± 3.6	0.239
Cre (mg/dL)	0.91 ± 0.19	0.91 ± 0.17	0.930	0.66 ± 0.13	0.67 ± 0.17	0.818
CysC (mg/L)	0.99 ± 0.22	1.04 ± 0.20	0.276	0.89 ± 0.16	0.91 ± 0.21	0.418
Cre/CysC	0.93 ± 0.13	0.88 ± 0.14	0.162	0.75 ± 0.10	0.74 ± 0.11	0.426
eGFRcre (mL/min/1.73 m^2^)	66.2 ± 14.0	64.6 ± 12.0	0.613	68.5 ± 12.9	69.1 ± 14.9	0.765
eGFRcys (mL/min/1.73 m^2^)	74.1 ± 15.9	62.4 ± 14.3	0.111	76.5 ± 14.0	75.2 ± 16.8	0.615
eGFRcys/eGFRcre	1.13 ± 0.18	1.07 ± 0.19	0.158	1.13 ± 0.17	1.10 ± 0.17	0.273
Hemoglobin (g/dL)	14.3 ± 1.5	14.0 ± 1.1	0.282	13.2 ± 1.0	13.2 ± 1.1	0.715
Albumin (g/dL)	4.3 ± 0.3	4.2 ± 0.3	0.289	4.4 ± 0.2	4.3 ± 0.3	0.807
Number of teeth, n	21.2 ± 8.7	19.2 ± 9.0	0.304	21.4 ± 8.2	19.9 ± 8.3	0.247
Tongue pressure (kPa)	35.6 ± 9.2	32.6 ± 7.4	0.141	33.3 ± 8.3	34.0 ± 9.3	0.635
Low articulatory oral motor skills (times/s)	5.86 ± 1.11	5.75 ± 1.13	0.658	6.22 ± 0.82	6.02 ± 0.74	0.127
**Item of oral frailty**						
Fewer teeth, n(%)	22(28.2)	14(51.9)	**0.035**	51(28.8)	16(34.0)	0.480
Difficulty in chewing, n(%)	12(15.4)	6(22.2)	0.554	28(15.8)	14(29.8)	**0.036**
Difficulty in swallowing, n(%)	16(20.5)	8(29.6)	0.425	43(24.3)	20(43.6)	**0.018**
Dry mouth, n(%)	17(21.8)	8(29.6)	0.438	52(29.4)	18(38.3)	0.076
Low articulatory oral motor skills, n(%)	34(43.6)	13(48.1)	0.823	58(32.8)	16(34.0)	0.863
**OF-5 score**	**1.29 ± 1.01**	**1.81 ± 1.00**	**0.023**	**1.31 ± 1.17**	**1.79 ± 1.37**	**0.017**
0, n(%)	17(21.8)	4(14.8)	0.580	53(29.9)	9(19.1)	0.199
1, n(%)	32(41.0)	4(14.8)	**0.018**	54(30.5)	15(31.9)	0.860
2, n(%)	21(26.9)	12(44.4)	0.099	40(22.6)	7(14.9)	0.315
3, n(%)	5(6.4)	7(25.9)	**0.011**	24(13.6)	9(19.1)	0.357
4, n(%)	3(3.8)	0(0.0)	0.568	4(2.3)	7(14.9)	**0.002**
5, n(%)	0(0.0)	0(0.0)	1.000	2(1.1)	0(0.0)	1.000
**Oral frailty states**						
Oral non-frailty, 0–1 OF-5 score, n(%)	49(62.8)	8(29.6)	**0.004**	107(60.5)	24(51.0)	0.250
Oral frailty, ≥2 OF-5 score, n(%)	29(37.2)	19(70.4)		70(39.5)	23(49.0)	

Cre, creatinine; CysC, cystatin C; eGFRcys, cystatin-based estimated glomerular filtration rate; eGFRcre, creatinine-based estimated glomerular filtration rate; OF-5, oral frailty five-item checklist.

**Table 5 nutrients-17-00017-t005:** Regression analysis.

Variables	Men		Women	
	OR (95%CI)	*p* Value	OR (95%CI)	*p* Value
**A.** Univariate logistic regression analysis for baseline factors predicting worsening frailty according to sex				
Age (per 1SD)	1.43(0.92–2.22)	0.108	1.28(0.93–1.75)	0.129
Body mass index (per 1SD)	1.18(0.77–1.82)	0.450	1.30(0.94–1.78)	0.108
Skeletal muscle mass index (SMI) (per 1SD)	0.78(0.49–1.22)	0.268	1.05(0.76–1.44)	0.783
Grip power (per 1SD)	**0.57(0.34–0.95)**	**0.022**	0.82(0.58–1.14)	0.232
Normal gait speed (per 1SD)	0.76(0.49–1.18)	0.213	0.97(0.70–1.33)	0.837
Cre/CysC (per 1SD)	0.71(0.44–1.15)	0.149	0.87(0.63–1.22)	0.420
eGFRcys/eGFRcre (per 1SD)	0.71(0.44–1.15)	0.145	0.83(0.59–1.16)	0.265
Hemoglobin (per 1SD)	0.79(0.52–1.21)	0.285	0.94(0.68–1.30)	0.714
Albumin (per 1SD)	0.79(0.51–1.22)	0.287	0.96(0.70–1.33)	0.806
Number of teeth (per 1SD)	0.80(0.53–1.22)	0.307	0.84(0.62–1.13)	0.255
Tongue pressure (per 1SD)	0.72(0.46–1.12)	0.139	1.08(0.78–1.50)	0.632
Low articulatory oral motor skills (per 1SD)	0.91(0.59–1.40)	0.657	0.78(0.57–1.07)	0.128
**Item of oral frailty**				
Fewer teeth (absence = 0, presence = 1)	**2.74(1.11–6.75)**	**0.028**	1.28(0.64–2.53)	0.487
Difficulty in chewing (absence = 0, presence = 1)	1.57(0.53–4.70)	0.427	**2.26(1.07–4.75)**	**0.037**
Difficulty in swallowing (absence = 0, presence = 1)	1.63(0.60–4.40)	0.340	**2.31(1.18–4.52)**	**0.015**
Dry mouth (absence = 0, presence = 1)	1.51(0.56–4.05)	0.418	1.49(0.76–2.92)	0.297
Low articulatory oral motor skills (absence = 0, presence = 1)	1.20(0.50–2.89)	0.682	1.06(0.54–2.09)	0.869
**OF-5 score** (per 1 point)	**1.65(1.06–2.57)**	**0.023**	**1.36(1.05–1.76)**	**0.019**
**Oral frailty states**				
Oral frailty, ≥2 OF-5 score (absence = 0, presence = 1)	**4.01(1.56–10.33)**	**0.004**	1.46(0.77–2.80)	0.248
**B.** Multivariate logistic regression analysis for baseline factors associated with worsening of frailty according to sex				
Age (per 1SD)	0.98(0.56–1.70)	0.934	1.15(0.81–1.64)	0.436
Body mass index (per 1SD)	1.53(0.82–2.84)	0.172	1.39(0.88–2.20)	0.156
Grip power (per 1SD)	0.62(0.34–1.13)	0.106	0.96(0.64–1.42)	0.820
Normal gait speed (per 1SD)	0.86(0.53–1.42)	0.564	1.13(0.80–1.62)	0.479
Skeletal muscle mass index (SMI) (per 1SD)	0.82(0.39–1.71)	0.592	0.86(0.54–1.37)	0.536
Cre/CysC (per 1SD)	0.91(0.53–1.58)	0.740	1.03(0.72–1.49)	0.863
**OF-5 score** (per 1 point)	1.49(0.91–2.45)	0.107	**1.32(1.00–1.75)**	**0.049**
**C.** Multivariate logistic regression analysis for baseline factors associated with worsening of frailty in men				
Age (per 1SD)	0.96(0.55–1.69)	0.894		
Body mass index (per 1SD)	1.50(0.80–2.82)	0.209		
Grip power (per 1SD)	0.61(0.33–1.11)	0.097		
Normal gait speed (per 1SD)	0.87(0.52–1.44)	0.582		
Skeletal muscle mass index (SMI) (per 1SD)	0.83(0.39–1.75)	0.622		
Cre/CysC (per 1SD)	0.94(0.54–1.64)	0.830		
**Oral frailty states**				
Oral frailty, ≥2 OF-5 score (absence = 0, presence = 1)	**3.38(1.23–9.28)**	**0.018**		

OR, odds ratio; CI, confidence interval; SD, standard deviation; Cre, creatinine; CysC, cystatin C; eGFRcys, cystatin-based estimated glomerular filtration rate; eGFRcre, creatinine-based estimated glomerular filtration rate; OF-5, oral frailty five-item checklist.

## Data Availability

Data supporting the findings of this study are available from the corresponding author upon reasonable request. However, the data are not publicly available due to privacy and ethical restrictions.

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
