# Peer review of "Oral Frailty and Its Relationship with Physical Frailty in Older Adults: A Longitudinal Study Using the Oral Frailty Five-Item Checklist"

_nutrients, 2024, doi:10.3390/nu17010017_

Round 1

Reviewer 1 Report

Comments and Suggestions for Authors

Thank you for this well written and very interesting study! This reviewer only has a few editorial recommendations for change.

line 24: change to 2 sentences as follows: "... articulatory oral skills. Limited studies exist."

line 27: "... in the Sasayama-Tambia Area study, Hyogo, Japan, and their..."

lines 39- 40: "...and is described as an age-related decrease in oral function. Oral frailty is further defined as..."

line 44: " delete the word "Older" and begin the sentence: " Signs of oral frailty..."

line 57: "... these advancements, the longitudinal..."

Tables1-3: when the table in part is on a second page, please add the headers above the table columns for clarity with the abbreviation "cont." after the title. For example for Table 1 add: Total, Men, Women, p 

Table 5: Please change the main title to simply read "Regression Analyses" without any more detail. Then add the A, B, and C. titles above each table after the letter.

line 324: please change the word pronounce to "pronounced" 

Thank you again for this interesting study!

Author Response

Reviewer 1

Thank you for this well written and very interesting study! This reviewer only has a few editorial recommendations for change.

line 24: change to 2 sentences as follows: "... articulatory oral skills. Limited studies exist."

line 27: "... in the Sasayama-Tambia Area study, Hyogo, Japan, and their..."

lines 39- 40: "...and is described as an age-related decrease in oral function. Oral frailty is further defined as..."

line 44: " delete the word "Older" and begin the sentence: " Signs of oral frailty..."

line 57: "... these advancements, the longitudinal..."

Tables1-3: when the table in part is on a second page, please add the headers above the table columns for clarity with the abbreviation "cont." after the title. For example for Table 1 add: Total, Men, Women, p

Table 5: Please change the main title to simply read "Regression Analyses" without any more detail. Then add the A, B, and C. titles above each table after the letter.

line 324: please change the word pronounce to "pronounced"

Thank you again for this interesting study!

Response to Reviewer Comments

We sincerely thank the reviewer for their thoughtful and constructive feedback. Below are our responses to each of the comments:

line 24: change to 2 sentences as follows: "... articulatory oral skills. Limited studies exist."

line 27: "... in the Sasayama-Tambia Area study, Hyogo, Japan, and their..."

lines 39- 40: "...and is described as an age-related decrease in oral function. Oral frailty is further defined as..."

line 44: " delete the word "Older" and begin the sentence: " Signs of oral frailty..."

line 57: "... these advancements, the longitudinal..."

line 324: please change the word pronounce to "pronounced"

  1. We have corrected the problem according to your suggestion. Please confirm.

Tables1-3: when the table in part is on a second page, please add the headers above the table columns for clarity with the abbreviation "cont." after the title. For example for Table 1 add: Total, Men, Women, p

Table 5: Please change the main title to simply read "Regression Analyses" without any more detail. Then add the A, B, and C. titles above each table after the letter.

  1. Please confirm that we have corrected the Tables according to your suggestion.

Reviewer 2 Report

Comments and Suggestions for Authors

The manuscript by Kusunoki et al. reports on their cross-sectional and longitudinal follow-up studies on oral frailty in community-dwelling older adults from the Sasayama-Tamba area in Japan. The study subject is relevant, and the text is very well written. The presented data suggest that oral frailty may serve as an early detection marker for the general frailty progression in older adults. Although supported by a comprehensive set of data, the outcomes are not surprising in the clinical field.

They authors are kindly asked to resolve two major issues to improve the quality of their report.

1)    The authors have pointed to sarcopenia in different sections without providing relevant information on its assessment. They should consider providing a paragraph in the introduction section on the relevance of sarcopenia in the progression of frailty. What was the prevalence of sarcopenia in their study? It is also important providing the assessment tools they used for the diagnosis of sarcopenia. Did they detect any correlations between sarcopenia and the progression of frailty or oral frailty? If not, please provide the results in supplemental materials.

The follow-up study needs to be better explained. Did the participants receive a treatment after the first survey? Because older adults are often under medical treatments, it is important to evaluate their effects on the study outcomes. Which type of training or treatments, how long and the type of medications? Can the authors explain why the frailly scores were improved at the second survey in so many participants?

Author Response

Reviewer 2

 The manuscript by Kusunoki et al. reports on their cross-sectional and longitudinal follow-up studies on oral frailty in community-dwelling older adults from the Sasayama-Tamba area in Japan. The study subject is relevant, and the text is very well written. The presented data suggest that oral frailty may serve as an early detection marker for the general frailty progression in older adults. Although supported by a comprehensive set of data, the outcomes are not surprising in the clinical field. They authors are kindly asked to resolve two major issues to improve the quality of their report.

1) The authors have pointed to sarcopenia in different sections without providing relevant information on its assessment. They should consider providing a paragraph in the introduction section on the relevance of sarcopenia in the progression of frailty. What was the prevalence of sarcopenia in their study? It is also important providing the assessment tools they used for the diagnosis of sarcopenia. Did they detect any correlations between sarcopenia and the progression of frailty or oral frailty? If not, please provide the results in supplemental materials.

2) The follow-up study needs to be better explained. Did the participants receive a treatment after the first survey? Because older adults are often under medical treatments, it is important to evaluate their effects on the study outcomes. Which type of training or treatments, how long and the type of medications? Can the authors explain why the frailly scores were improved at the second survey in so many participants?

Response to Reviewer Comments

We thank the reviewer for their detailed and insightful feedback, which has helped us improve the clarity and comprehensiveness of our manuscript. Below, we provided responses and describe the revisions made in response to each suggestion.

1) The authors have pointed to sarcopenia in different sections without providing relevant information on its assessment. They should consider providing a paragraph in the introduction section on the relevance of sarcopenia in the progression of frailty. What was the prevalence of sarcopenia in their study? It is also important providing the assessment tools they used for the diagnosis of sarcopenia. Did they detect any correlations between sarcopenia and the progression of frailty or oral frailty? If not, please provide the results in supplemental materials.

  1. Thank you for your point regarding sarcopenia. The AWGS criteria are widely used for diagnosing sarcopenia. In a cross-sectional study, the prevalence of sarcopenia was calculated based on the AWGS criteria and found to be approximately 7% in both men and women. The following statement has been added:

“A typical phenotype of physical frailty is age-related muscle loss, known as sarcopenia. Using the AWGS2019 diagnostic criteria for Asians, the prevalence of sarcopenia was estimated to be approximately 7% in both men and women, with no significant gender differences.” (Lines 234-237)

“The prevalence of sarcopenia was observed to be higher in both men and women with oral frailty. However, the difference was not statistically significant in men. Overall, the findings suggest that sarcopenia exhibits a similar trend to physical frailty.” (Lines 241-244)

We also added a description of the measurement method for sarcopenia based on the AWGS2019 criteria.

2.4. Diagnosis of sarcopenia

Sarcopenia was defined according to the criteria for the Asia Working Group for Sarcopenia (AWGS) 2019 [30]. Body composition was evaluated by bioelectrical impedance analysis (BIA) using an InBody 770® (InBody Japan Inc., Tokyo, Japan). The skeletal muscle mass index (SMI) was calculated as SMM/height2 (kg/m2). The handgrip power, and the normal and maximal gait speed, 5-time chair stand test (5CS), Timed Up and Go test (TUG), and Short Physical Performance Battery (SPPB) scores were evaluated as described previously [30]. Sarcopenia was considered if the participants had a low SMI (< 7.0 kg/m2 in men; < 5.7 kg/m2 in women) and weak handgrip strength (< 28 kg in men; < 18 kg in women) or low physical performance (normal gait speed < 1.0 m/s, 5CS ≥ 12 s or SPPB ≤ 9).” (Lines 182-192)

2) The follow-up study needs to be better explained. Did the participants receive a treatment after the first survey? Because older adults are often under medical treatments, it is important to evaluate their effects on the study outcomes. Which type of training or treatments, how long and the type of medications? Can the authors explain why the frailly scores were improved at the second survey in so many participants?

Thank you for your insightful feedback. We have added following sentences in the discussion.

“Approximately one-fourth of the participants demonstrated improvements in J-CHS scores for both men and women during the Second Survey. Neither pharmacological nor exercise interventions specifically targeting frailty were implemented. Therefore, the observed improvement in J-CHS scores may be attributed to increased activity levels among older adults following the lifting of COVID-19-related restrictions in Japan, such as the state of emergency declarations.” (Lines 356-361)

Reviewer 3 Report

Comments and Suggestions for Authors

This paper presents an evaluation of the relationship of an oral frailty measure to a physical frailty measure. Overall, the technical portions of the paper are clearly and appropriately presented.  However, there is little context for this analysis or discussion of the implications of this study.  The value of the measures used, either for practice or research, is not discussed. There is also very little about the importance of oral health or of assessing it.  Yet oral health can have a significant impact on the individual's health and well-being. In the Discussion section there is an example of how the lack of a conceptual foundation leaves unanswered questions.  The authors state that oral dysfunction may be a marker of systemic health decline.  However, the reader may well ask if it is just a marker or is it an important contributor to systemic decline? This omission significantly reduces the value of the paper but is easily remedied. One additional, minor, point:  Table 1 needs a note indicating the comparison for which the p value was generated.

Author Response

Reviewer 3

This paper presents an evaluation of the relationship of an oral frailty measure to a physical frailty measure. Overall, the technical portions of the paper are clearly and appropriately presented.  However, there is little context for this analysis or discussion of the implications of this study.  The value of the measures used, either for practice or research, is not discussed. There is also very little about the importance of oral health or of assessing it.  Yet oral health can have a significant impact on the individual's health and well-being. In the Discussion section there is an example of how the lack of a conceptual foundation leaves unanswered questions.  The authors state that oral dysfunction may be a marker of systemic health decline.  However, the reader may well ask if it is just a marker or is it an important contributor to systemic decline? This omission significantly reduces the value of the paper but is easily remedied. One additional, minor, point:

Response to Reviewer Comments

We sincerely thank the reviewer for their thoughtful and constructive feedback. Below are our responses to each of the comments. We consider oral dysfunction to be an important contributor to systemic decline.

  1. We have added the following sentence.

“Oral health is a critical component of overall health, life satisfaction, quality of life, and self-perception. Impairment of oral function is highly prevalent among older adults, and aging has been reported to interact indirectly with various domains of frailty through multiple pathways. A clear example of this relationship is age-related functional oral deterioration, characterized by poor dental hygiene, inadequate dental prostheses, and dietary deficiencies, which collectively contribute to an increased risk of nutritional frailty [1].

Oral frailty is defined as an age-related gradual decline in oral function, often accompanied by deteriorations in physical functions. This condition is associated with significant adverse health outcomes in older adults, including increased mortality, physical frailty, functional disabilities, reduced quality of life, and a higher risk of hospitalization and falls [2]. Poor oral health in the elderly is a major health concern due to its links to the pathogenesis of systemic frailty, suggesting that it is a multidimensional geriatric syndrome. As such, oral frailty may serve as a potential risk factor for systemic frailty [3].” (Lines 39-52)

“Oral dysfunction is regarded as a significant contributor to systemic decline. Oral frailty is defined as a mild decline in oral functions during the early and reversible stages of frailty. Many community-dwelling older people have reduced oral function or oral hypofunction, which is significantly associated with frailty and aging. Appropriate evaluation of oral function and effective intervention to suppress oral function deterioration may be effective in extending the healthy life expectancy of older people [35].” (Lines 314-320)

Table 1 needs a note indicating the comparison for which the p value was generated.

We have added the following sentence.

“The p-value represents the significant difference between women and men.” (Line 251)